# Iodine-Loaded Calcium Titanate for Bone Repair with Sustainable Antibacterial Activity Prepared by Solution and Heat Treatment

**DOI:** 10.3390/nano11092199

**Published:** 2021-08-26

**Authors:** Seiji Yamaguchi, Phuc Thi Minh Le, Seine A. Shintani, Hiroaki Takadama, Morihiro Ito, Sara Ferraris, Silvia Spriano

**Affiliations:** 1Department of Biomedical Sciences, College of Life and Health Sciences, Chubu University, 1200 Matsumoto, Kasugai 487-8501, Aichi, Japan; minhphuc@isc.chubu.ac.jp (P.T.M.L.); shintani@isc.chubu.ac.jp (S.A.S.); takadama@isc.chubu.ac.jp (H.T.); m-ito@isc.chubu.ac.jp (M.I.); 2Politecnico di Torino, Corso Duca degli Abruzzi 24, Corso Duca degli Abruzzi 24, 10129 Torino, Italy; sara.ferraris@polito.it (S.F.); d002307@polito.it (S.S.)

**Keywords:** iodine, calcium titanate, titanium, apatite-forming ability, cytotoxicity

## Abstract

In the orthopedic and dental fields, simultaneously conferring titanium (Ti) and its alloy implants with antibacterial and bone-bonding capabilities is an outstanding challenge. In the present study, we developed a novel combined solution and heat treatment that controllably incorporates 0.7% to 10.5% of iodine into Ti and its alloys by ion exchange with calcium ions in a bioactive calcium titanate. The treated metals formed iodine-containing calcium-deficient calcium titanate with abundant Ti-OH groups on their surfaces. High-resolution XPS analysis revealed that the incorporated iodine ions were mainly positively charged. The surface treatment also induced a shift in the isoelectric point toward a higher pH, which indicated a prevalence of basic surface functionalities. The Ti loaded with 8.6% iodine slowly released 5.6 ppm of iodine over 90 days and exhibited strong antibacterial activity (reduction rate >99%) against methicillin-resistant *Staphylococcus aureus* (MRSA), *S. aureus*, *Escherichia coli*, and *S. epidermidis*. A long-term stability test of the antibacterial activity on MRSA showed that the treated Ti maintained a >99% reduction until 3 months, and then it gradually decreased after 6 months (to a 97.3% reduction). There was no cytotoxicity in MC3T3-E1 or L929 cells, whereas apatite formed on the treated metal in a simulated body fluid within 3 days. It is expected that the iodine-carrying Ti and its alloys will be particularly useful for orthopedic and dental implants since they reliably bond to bone and prevent infection owing to their apatite formation, cytocompatibility, and sustainable antibacterial activity.

## 1. Introduction

Since the observation of the direct contact of titanium metal (Ti) to bone under optical microscopy (osseointegration) in the 1960s, Ti and its alloys have been widely used as bone repair materials for cases in which large loads are involved, such as artificial joints and artificial vertebral bodies. However, the loosening of these implants has often been reported since this type of machined Ti has poor bone bonding and often requires a long period of time to achieve osseointegration [1,2].

To impart bone-bonding capacity to Ti and its alloys, various surface modifications, including plasma spray-coating, sputtering, and sol-gel or alternative soaking, have been used to provide a coat of apatite or other calcium phosphates [3,4]. Alkaline or acidic solution and heat [5,6], blasting [7], hydrothermal treatment [8], ion implantation [9], and anodic oxidation [10] have also been examined. These studies revealed a roughened surface on the micro- and nanometer scales, and their hierarchical arrangement enhanced preosteoblast cell activity, including proliferation, differentiation, and gene expression [11,12,13,14]. An apatite coating or providing Ti and its alloys with apatite formation capacity induced direct bone bonding in a short period of time [15,16].

Ti with a moderately porous structure enables blood and cells to enter into the porous body, which results in bone ingrowth [17]. A number of studies have investigated the effect of pore structure on bone ingrowth, reporting that porous Ti with fully interconnected pores in the range of 300–1100 μm exhibited a reasonable amount of bone ingrowth despite differences in porosity [18,19,20]. By contrast, the elastic modulus and mechanical strength are highly reliant on the porosity [21]. Recent advances in manufacturing techniques, including selective later melting and electron beam melting, have attracted a great deal of attention due to their ability to control the internal structure of porous Ti [22,23].

Among the surface modifications described above, heat treatment and solution are both cost-effective and stimulate the formation of a uniform bioactive surface layer, even on the inner surface of a porous body [24,25]. Remarkable increases in bone ingrowth and ectopic bone formation in muscle were observed when a porous Ti implant subjected to NaOH, water, and heat treatment was implanted into a femoral condyle of a rabbit and a dorsal muscle of a beagle dog, respectively [25].

Despite the successful results of increased bone bonding and bone ingrowth obtained with solution and heat treatment, the challenge remains of providing Ti with antibacterial activity. It was reported that typically 1–2% of patients with total hip arthroplasties suffer deep infections [26], 2% following spinal operations, and 13% for endoprosthetic replacement of large bone defects after tumor resection, while the percentage can reach up to 70% in external fixation cases [27,28]. Dental peri-implant disease and infection have become a main focus of oral implantology [29].

To obtain antibacterial activity in addition to bioactivity, silver (Ag) has been incorporated into Ti and Ti alloy surfaces by modified solution and heat treatments in which Ti is soaked in a H_2_O_2_/AgNO_3_ mixed solution [30], in a NaOH/AgC_2_H_3_O_2_ mixed solution [31], or in a AgNO_3_ solution following NaOH-CaCl_2_+SrCl_2_ heat treatment [32,33,34]. Although the treated Ti displayed strong antibacterial activity against *Staphylococcus aureus*, *Escherichia coli*, and so on, in vitro, its in vivo antibacterial activity was insufficient [34] because only a low amount of Ag could be incorporated into the Ti surface due to the cytotoxicity of Ag. The range of Ag on Ti and Ti alloys that exhibits antibacterial activity without cytotoxicity is narrow, so accurate control of Ag release is required to achieve a proper balance between antibacterial activity and cytocompatibility [35].

Iodine is an alternative candidate that exhibits a broad antibacterial spectrum and has been used as a skin disinfectant for a long time [36]. In other aspects, iodine is an essential element in the living body and is utilized in catheter angiograms at high concentrations (300 mg I/mL and above) in vivo [37], demonstrating the cytocompatibility of iodine. A few reports demonstrate iodine being introduced into metal surfaces. Takaya et al. reported that 0.1 mass% of iodine could be loaded on aluminum by electrodeposition following anodization, and the treated metal exhibited strong antibacterial activity (reduction >99%) against *S. aureus* and *E. coli* [38]. Tsuchiya et al. [39] and Shirai et al. [40] showed that Ti and Ti-6Al-4V alloy loaded with iodine through a similar method exhibited strong antibacterial activity. A clinical trial of the treated alloy reported remission of infection and prevention of postoperative infections without any obvious cytotoxicity. However, electrodeposition and anodization are not commonly applied to the inner surface of porous Ti due to their electric shielding effect [41,42].

In this paper, we report a novel solution and heat treatment in which a large amount of iodine can be incorporated into the calcium titanate formed on Ti and its alloys, which exhibit a marked bone-bonding ability [43]. The in vitro bioactivity, antibacterial activity, and cytotoxicity were evaluated in terms of apatite formation, the antibacterial activity value, and cell proliferation on the treated metals. The sustainable antibacterial activity of the treated metal was evaluated by soaking the metal in phosphate-buffered saline for up to six months.

## 2. Materials and Methods

### 2.1. Materials Resources

A commercially pure, medical-grade Ti plate (ISO5832-2) (Ti > 99.5%) was provided by Nilaco Co., Tokyo, Japan. Ti-6Al-4V alloy plate (Ti = balance, Al = 6.18, V = 4.27 mass%) and Ti-15Zr-4Nb-4Ta alloy plate (Ti = balance, Zr = 14.51, Nb = 3.83, Ta = 3.94, Pd = 0.16, and O = 0.25 mass%) were supplied by Kobelco Research Institute, Inc., Hyogo, Japan.

The chemical reagents (NaOH, CaCl_2_, ICl_3_, ICl, NaI and povidone iodine (PVP-I)) used for surface treatment were reagent-grade and purchased from Kanto Chemical Co., Inc., Tokyo, Japan. Reagent-grade NaCl, NaHCO_3_, KCl, K_2_HPO_4_·3H_2_O, MgCl_2_·6H_2_O, CaCl_2_, Na_2_SO_4_, tris-hydroxymethylaminomethane (CH_2_OH)_3_CNH_2_, and 1 M HCl were purchased from Nacalai Tesque, Inc., Kyoto, Japan, and used for the preparation of simulated body fluid (SBF). Minimum essential medium (MEM) was obtained from Gibco, Thermo Fisher Scientific, Waltham, MA, USA, and used for the cell culture test. Nutrient broth (Eiken Chemical Co., Ltd. Tochigi, Japan) and RPMI 1640 (Fujifilm Wako Pure Chemical Corp., Osaka, Japan) were used for the antibacterial activity test.

### 2.2. Surface Treatment

Titanium and its alloy plates were cut into a rectangular shape (10 × 10 × 1 mm^3^), and ground with #400 diamond plates. They were cleaned with acetone, 2-propanol, and ultrapure water in an ultrasonic cleaner for 30 min each, then dried at 40 °C in an incubator overnight. 

A calcium titanate surface was formed on Ti and its alloy samples as the result of NaOH-CaCl_2_ heat treatment, as in our previous report [44]. Briefly, the samples were initially soaked in 5 mol/L NaOH aqueous solution at 60 °C for 24 h, and subsequently in 100 m mol/L CaCl_2_ at 40 °C for 24 h with shaking in an oil bath. The samples were subsequently heated to 600 °C at a rate of 5 °C/min and maintained at 600 °C for 1 h, followed by natural cooling in an Fe-Cr electrical furnace. Some of the treated samples were ultimately soaked in iodine-containing solutions of 10 or 100 m mol/L ICl_3_, 10% ICl at a temperature of 40, 60, or 80 °C, i.e., the solutions that produce positively charged iodine ions. The other samples were soaked in 100 m mol/L of NaI or 1000 ppm PVP-I at 60 °C, which produce negatively charged iodine ions and/or isolated iodine. The notations of the samples used in this study are listed in Table 1.

### 2.3. Surface Characteristics

#### 2.3.1. Scanning Electron Microscopy

The surface and cross-sectional area of the prepared Ti and its alloy samples soaked in SBF were coated with a Pt/Pd thin film and observed under a field-emission scanning electron microscope (FE-SEM, S-4300, Hitachi Co., Tokyo, Japan) at an acceleration voltage of 15 kV. For the cross-sectional SEM observation, the metal samples were bent to produce microcracks on their surfaces, and the cross-sections of these cracks were observed as reported in our previous study [45].

#### 2.3.2. X-ray Photoelectron Spectroscopy

The compositional elements of the surface of the treated Ti and alloys samples that were subjected to the final ICl_3_ solution treatment were analyzed using X-ray photoelectron spectroscopy (XPS, PHI 5000 Versaprobe II, ULVAC-PHI, Inc., Kanagawa, Japan), with an Al-Kα radiation line as the X-ray source at a take-off angle of 45°. The binding energy of the measured spectra was calibrated by referencing the C1s peak of the surfactant CH_2_ groups on the substrate occurring at 284.8 eV.

#### 2.3.3. Thin-Film X-ray Diffraction

The crystalline phase of the samples’ surfaces was analyzed using TF-XRD (Model RNT-2500, Rigaku Co., Tokyo, Japan) employing a CuKα X-ray source operating at 50 kV and 200 mA. The glancing angle of the incident beam was set to an angle of 1° against the sample surface.

#### 2.3.4. Scratch Resistance Measurements

The scratch resistance of the surface layer that formed on the Ti and alloy samples subjected to the final ICl_3_ or ICl solution treatment was measured using a thin-film scratch tester (Model CSR-2000, Rhesca Co., Tokyo, Japan). A stylus with a diameter of 5 μm and a spring constant of 200 g/mm was employed and pressed onto the samples with a loading rate of 100 mN/min, amplitude of 100 μm, and scratch speed of 10 μm/s, based on the data in the JIS R-3255 standard. Five areas were measured for each sample, and their average value was used.

#### 2.3.5. Zeta Potential Measurements

The zeta potential titration curve for both the untreated Ti samples and those subjected to the final ICl_3_ solution was obtained as a function of pH using an electrokinetic analyzer (SurPASS, Anton Paar GmbH, Graz, Austria) equipped with an adjustable gap cell. A pair of samples was mounted in the cell to form a gap between the two surfaces adjusted to approximately 100 μm. During the measurement, the electrolyte (KCl 0.001 mol/L) flow (approximately 100 mL/min) through the gap and the streaming potential was measured, and the surface zeta potential was calculated by the instrument’s software [46,47]. The pH of the electrolyte (which starts at approximately 5.5) was varied as the result of the addition of 0.05 M HCl (acid titration curve) or 0.05 M NaOH (basic titration curve) through the instrument’s automatic titration unit. For each pH value, 4 measurements were recorded and the average value was calculated along with the standard deviation. Two separate sample pairs were used for the acidic and basic titrations, and washing of the instrument (2 cycles with 500 mL water) was performed between the two measurements. The isoelectric point (IEP) was determined as the intersection of the titration curve with the horizontal axis (zeta potential equal to 0 mV). To examine the initial reaction of the treated sample in SBF, the zeta potential at physiological pH was also monitored on an independent set of samples as a function of the elapsed time up to 1 h using diluted SBF (pH = 7, at 20 mS/m) as the electrolyte and no titration.

#### 2.3.6. Ion Release

The Ti samples subjected to the final ICl_3_ solution treatment were immersed in 2 mL of PBS with gentle shaking at a speed of 50 strokes/min at 36.5 °C. The iodine ion concentrations released from the treated samples after determined periods of soaking for up to 90 days were measured by inductively coupled plasma emission spectroscopy (ICP, SPS3100, Seiko Instruments Inc., Chiba, Japan). The measurement was repeated 3 times for independently prepared samples, and their averaged values were calculated.

### 2.4. Apatite Formation

The Ti and alloy samples subjected to the final ICl_3_ or ICl solution treatment were immersed in 24 mL of pre-warmed SBF with ion concentrations nearly equal to those of human blood plasma for 3 days at 36.5 °C [15]. After removal from the SBF, apatite formation on the sample surface was observed under FE-SEM. Their crystalline structure was examined by TF-XRD.

### 2.5. Cell Proliferation

MC3T3-E1 mouse preosteoblast cells (ATCC, Manassas, VA, USA) and L929 mouse fibroblast cells (DS Pharma Biomedical Co., Ltd., Osaka, Japan) were cultured at 37 °C, 5% CO_2_, in MEM supplemented with 10% (*v*/*v*) fetal bovine serum, 1% (*v*/*v*) penicillin, and streptomycin. Cells were subcultured after 2–3 days when the confluency reached 70–80%. These cells were seeded on the untreated and the treated Ti (ϕ18× 1 mm) that had been subjected to the final ICl3 solution treatment in a 12-well plate at a density of 105 cells/mL, and then cultured in MEM at 37 °C, 5% CO2. After the culture periods of 1, 3, and 7 days, the cell proliferation on the samples was evaluated by adding the cell count reagent SF (Nacalai Tesque, Kyoto, Japan) followed by incubation for 2 h. The formazan product after the incubation was measured at 450 nm using a microplate reader (iMarkTM, Bio-Rad, Hercules, CA, USA). All samples were sterilized with ethylene oxide gas before the cell experiments. The experiment was repeated four times.

### 2.6. Antibacterial Activity Test

The antibacterial activity of the treated Ti against methicillin-resistant *Staphylococcus aureus* (MRSA; isolated from the nasal cavity of MRSA patient), *S. aureus* (ATCC, 6538P), *S. epidermidis* (ATCC, 49134), and *Escherichia coli* (*E. coli*; IFO3972) was evaluated according to the ISO22196 standard [48]. Bacterial cell suspensions of 100 μL in 1/500 nutrient broth were inoculated on 25 × 25 × 1 mm^3^ Ti samples that had been subjected to the final ICl_3_ solution treatment, then covered with a 20 × 20 mm^2^ flexible polypropylene film so that the suspensions were in close contact with the samples. The polypropylene film was previously sterilized with ethanol and dried for 7 days in a clean bench before use. The inoculated samples were placed in a 100 mm diameter Petri dish along with a sterilized plastic cap filled with sterilized pure water to prevent drying of the bacterial suspension, then stored in an incubator under 95% relative humidity at 35 °C for 24 h. Notably, RPMI 1640 medium with an abundance of nutrients was used for the suspension medium of *S. epidermidis* instead of the 1/500 nutrient broth that was used for other bacteria cells to maintain the cell number after incubation. After incubation, each sample was carefully washed with 10 mL of soybean casein digest broth containing lecithin and polyoxyethylene sorbitan monooleate (SCDLP broth) to collect the bacteria. The recovered suspension was subjected to 10-fold serial dilutions, then 1 mL of the recovered suspension with or without the dilution was placed in Petri dishes containing standard plate count agar at 35 °C for 48 h. After incubation, the number of viable bacteria cells was calculated using the dilution factor and the number of colonies that had been counted on the Petri dish. Finally, the antibacterial activity value (*R*) was calculated for each specimen as follows:*R* = {log(*B*/*A*) − log(*C*/*A*)} = log(*B*/*C*)(1)
where *A* is the number of viable *E. coli* recovered from the untreated specimen immediately after inoculation, *B* is the number of viable *E. coli* recovered from the untreated specimen immediately after 24 h incubation, and *C* is the number of viable *E. coli* recovered from the treated specimen immediately after 24 h incubation.

To evaluate the stability of the antibacterial activity, the treated samples were kept for one week in an incubator with a relative humidity of 95% at 80 °C (a moisture resistance test), or soaked in PBS for up to 6 months at 36.5 °C with gentle shaking at a speed of 50 strokes/min (a water resistance test). The former is an accelerated test of stability during storage while the latter mimics physiological conditions to allow an evaluation of the stability after implantation. After each test, the sample was evaluated and its antibacterial activity value was calculated.

### 2.7. Statistical Analysis

Comparison of the scratch resistance of the surface layer values between each sample was performed by Tukey’s test at the 5% significance level. The statistical test in the cell proliferation was performed as follows: The Kolmogorov–Smirnov test and the *F*-test confirmed the normality and homoscedasticity of the data. The data for which the null hypothesis of normality and homoscedasticity was not rejected were tested by Student’s *t*-test, and the data for which the null hypothesis of homoscedasticity was rejected were tested by Welch’s *t*-test. The significance level of the *p*-value was set to 5% in all statistical tests.

## 3. Results

### 3.1. The Surface Characteristics

Nanostructured surface layers composed of calcium titanate and rutile that were approximately 1 μm thick were produced on Ti and alloy samples by NaOH, CaCl_2_, and heat treatment, as reported in our previous study [43,44].

These samples were soaked in various kinds of iodine solutions and their chemical compositions were examined by XPS, as shown in Table 2. Table 2 shows that 0.7–10.5% of the iodine was incorporated into the Ti and alloy surface when the samples were soaked in ICl_3_ or ICl solution, whereas no iodine was detected on the samples that were soaked in PVP-I or NaI solution. A lower amount of calcium was detected on the samples with a higher iodine content except for the Ti sample treated with 100 mM ICl_3_ at 80 °C, which had both low calcium and iodine contents. No chlorine was detected in the samples treated with ICl_3_ and ICl, indicating no residual ICl_3_ or ICl solution on the sample surfaces after washing.

XPS narrow spectra of I 3d and Ca 2p were obtained on the Ti and its alloy samples (S1_Ti, S1_Ti64, and S1_Ti1544) with maximum iodine contents that were introduced by soaking in 10 mM ICl_3_ at 80 °C following NaOH-CaCl_2_ heat treatment. They exhibited two pairs of split I 3d peaks that were approximately 619 and 630 eV, and 623 and 635 eV in binding energy (Figure 1a). The former pair is attributed to the negatively charged iodine (NIST XPS data base Version 4.1: 7681-82-5), whereas the latter pair is attributed to the positively charged iodine (NIST XPS data base Version 4.1: 10450609). Interestingly, the positively charged iodine peak intensity gradually decreased in the course of the measurement process, while the negatively charged iodine peak intensity was the opposite (Appendix A). This indicates that a portion of the positively charged iodine transformed into negatively charged iodine under irradiation with an electron beam during the course of the measurement. The treated samples also showed split peaks of approximately 347 and 350 eV, which are attributed to the Ca 2p3/2 and Ca 2p1/2 of CaO, respectively (NIST XPS data base Version 4.1: 1305-78-8), and their intensities were less than that on Ti as NaOH-CaCl_2_ heat treatment (Figure 1b). The fitting profile of O1s shows that both the acidic and basic Ti-OH groups appeared abundantly on the treated Ti and alloys compared with those on Ti with NaOH-CaCl_2_ heat treatment (Figure 1c).

Figure 2 shows the surface and cross-sectional FE-SEM images of the samples treated with ICl_3_ or ICl solution. Surface layers approximately 1 μm thick remained on the Ti and alloy samples soaked in 10 mM ICl_3_ at 80 °C. By contrast, the thickness of the surface layer significantly decreased to approximately 0.1 μm when the Ti sample was soaked in 100 mM ICl_3_ at 80 °C (see S2_Ti). A slightly damaged morphology was observed for the Ti samples treated with 100 mM ICl_3_ at 60 and 40 °C, although the thickness of the surface layer remained at ca. 1 μm (S3_Ti and S4_Ti).

The critical scratch loads of the Ti and Ti-6Al-4V samples after being soaked in 10 mM ICl_3_ at 80 °C were 33.3 and 41.5 mN, respectively, whereas the Ti-15Zr-4Nb-4Ta sample exhibited a higher value of 119 mN, as shown in Table 3. Critical scratch loads of 25.3 to 27.3 mN were detected on the Ti samples treated with 100 mM ICl_3_ or 10% ICl solution, which are slightly lower than that of the Ti sample treated with 10 mM ICl_3_, although no statistically significant differences were found between them. A statistically significant difference was detected between these samples and the Ti-6Al-4V sample after being soaked in 10 mM ICl_3_ at 80 °C.

The thin-film X-ray diffraction profiles of these samples are shown in Figure 3. The profiles show that calcium titanate, such as CaTi_2_O_4_, CaTi_2_O_5_ and CaTi_4_O_9_ (JCPDS Powder Diffraction Data Files 00-026-0333, 01-072-1134, 00-025-1450), and rutile were detected on C_Ti, and the peak positions were essentially unchanged even after the ICl_3_ or ICl treatment, irrespective of the concentration and/or temperature of the solution, except slight shifts in the broad peaks at 25.4 and 48.6 degrees to those at 24.8 and 48.3 degrees, respectively. This indicates that iodine (probably accompanied by hydrogen due to the hydrolysis of iodic acid) was incorporated into the crystal structure of calcium titanate to form iodine-containing calcium-deficient forms of calcium titanate, such as I_x_H_y_Ca_1−(0.5x+0.5y)_Ti_4_O_9_, I_x_H_y_Ca_1−(0.5x+0.5y)_Ti_2_O_4_, and I_x_H_y_Ca_1−(0.5x+0.5y)_Ti_4_O_5_. A decreased intensity of the iodine-containing calcium-deficient calcium titanate was observed on the Ti samples soaked in 100 mM ICl_3_ at 80 °C, which is consistent with the FE-SEM image revealing a thinner surface layer.

Figure 4 shows the zeta potential titration curves of the untreated and treated Ti samples (S1_Ti). The isoelectric point (IEP) of the treated surface was approximately 5.3 (close to the pH value of the KCl electrolyte at the beginning of the acid/basic measurements). At this pH, the net electrokinetic charge of the surface becomes null.

Consequently, the surface is negatively charged at physiological pH (approximately −34 mV at pH = 7.4). The curve shows an evident acid plateau (with positive zeta potential values), attributable to the presence of basic (protonated) functionalities. The shift in the IEP toward the basic range with respect to an untreated Ti surface is in agreement with the formation of basic functional groups due to the surface treatment. The onset of the acidic plateau is at pH ≈ 4, indicating that the basic functionalities created on the surface act as a strong base (they are protonated at a relatively high pH). On the other hand, the plateau in the basic range has an onset at pH 8.5: the surface treatment creates some acidic groups, but they act as a weak acid and are deprotonated only at a very high pH. The standard deviation of the zeta potential is very small for the entire measurement process (the error bars are almost not visible in the graph), evidencing the good chemical stability [49] of the surface through the whole pH range explored, and with more stability compared with the untreated Ti surface. Comparing the obtained zeta potential titration curve with the XPS data, we can determine that, in the case of the treated surface, the acidic and basic functionalities are due to the OH groups and that the basic ones play a larger role in determining the surface charge of the treated material because of their chemical reactivity. A contribution by basic and acidic functionalities due to iodine (HOI as an amphoteric functional group) may also be supposed [50].

### 3.2. Apatite Formation

When the Ti and alloy samples treated with ICl_3_ or ICl solution were soaked in SBF, they formed numerous spherical particles on their surface within 3 days, irrespective of the concentration and/or temperature of the solution, whereas no particles were found on the Ti that was treated with NaOH-CaCl_2_-heat, as shown in Figure 5. These spherical particles were determined to be low crystalline apatite (JCPDS Powder Diffraction Data Files 00-055-0592) by thin-film X-ray diffraction, as shown in Figure 6, in which the broad peak around 32° in 2θ is attributed to the low crystallinity of apatite [51]. The initial reaction between the SBF and sample surface (S1_Ti) was monitored by measurement of the zeta potential in diluted SBF, as shown in Figure 7. Figure 7a shows that the first value of the zeta potential is close to the one obtained at physiological pH in KCl (ζ = −38 mV at pH 7). The surface zeta potential displayed a trend toward less-negative values during the course of the measurement (ζ = −33 mV after 1 h). The change in the pH of the solution was very limited (pH = 6.8 after 1 h, as shown in Figure 7b), but it did display a trend toward slightly acidic values. The standard deviation of the zeta potential values obtained in diluted SBF was very small in both cases once again. The characteristics of the Ti and its alloy samples used in this study are summarized in Table 4.

### 3.3. Ion Release

Figure 8 shows the iodine concentration (closed circle) released from the treated sample (S1_Ti) as a function of the square root of the soaking time in PBS. Figure 8 shows that the treated Ti initially released 4.0 ppm of iodine within 6 h and further slowly released another 1.6 ppm over 90 days. After the ion-release test, the sample was subjected to XPS analysis to reveal the remaining iodine on the metal. As a result, we found that 1.7 mass% remained on the surface after 90 days, indicating that 79.6% of the iodine was released during the test. The percentage of iodine released from S1_Ti was calculated from the iodine concentration in PBS and iodine amount remaining on the metal after the test (closed square).

### 3.4. Cell Proliferation

The MC3T3-E1 and L929 cell proliferation on the treated sample (S1_Ti) during the culture periods for up to 7 days of incubation time was assessed and compared with that of untreated Ti, as shown in Figure 9. At all of the culture period time points except 3 days, the MC3T3-E1 and L929 cell proliferation on the treated Ti was comparable to that on untreated Ti. On day 3, a significant increase in MC3T3-E1 and decrease in L929 was detected on the treated Ti.

### 3.5. Antibacterial Activity

The antibacterial activity and its short-term stability in the Ti samples treated with higher and lower iodine contents on their surface were evaluated according to the ISO 22196 standard: S1_Ti with 8.6% and S3_Ti with 2.3% of iodine. Figure 10 shows that both of the treated Ti samples exhibited no colony formation, indicating potent antibacterial activity (reduction rate > 99%). This potent antibacterial activity remained for 1 week for S1_Ti, even after the humidity or water resistance test for 1 week in which no or only a scant amount of colony formation occurred. By contrast, some amount of colony formation occurred for S3_Ti with 2.3% iodine after the humidity test, and a larger amount was observed after the water resistance test, although these amounts were still lower than those for the untreated Ti.

The antibacterial activity values of S1_Ti with or without the humidity and water resistance test against MRSA are summarized in Table 5. The value reached as high as 4.3 after treatment and being subjected to the humidity test, and remained over two (reduction rate > 99%) even after the water resistance test. A similar trend was observed for *S. aureus* and *E. coli*, as shown in Table 5 and Table 6. The treated Ti also displayed strong antibacterial activity against *S. epidermidis* (Table 6).

To examine the long-term stability of this antibacterial activity, the antibacterial activity values of S1_Ti were examined with the water resistance test up to 6 months. The values were found to be 3.5, 2.0, and 1.6 after 1, 3, and 6 months, respectively. This indicates that, although the antibacterial activity of the treated Ti gradually decreased with soaking duration in a simulated physiological environment, it remained at a level corresponding to a 97.3% reduction.

## 4. Discussion

The calcium titanate formed on the Ti and alloy samples by sequential treatment with NaOH, CaCl_2_, and heat creates a layered structure in which calcium ions exist in the TiO_6_ octahedral [52]. Thus, the treated metals exhibit bone bonding that is attributed to their high apatite formation capability when they are activated by the subsequent hot water treatment [43,44]. The correlation between apatite formation and bone bonding was initially reviewed by Kokubo et al. in 2006 [15]. They described the quantitative correlation of apatite formation in SBF with in vivo bone bioactivity. They noted a few exceptional cases in which this criterion is not valid, e.g., if the materials are highly resorbable or cytotoxic. Fujibayashi et al. suggested that the materials able to form apatite within 3 days in SBF are recommended for practical use [53]. In 2014, Zadpoor reviewed 33 published studies available in the literature at the time, in all of which the in vitro apatite-forming ability and in vivo performance of two or more biomaterials were compared [16]. Zadpoor concluded that, in the majority of the studies (25/33), the SBF immersion test proved successful in predicting the relative in vivo bioactivity of the materials. We also showed that the calcium titanate formed on Ti and its alloys bond to living bone; some other functional metal ions such as strontium, magnesium, and silver have also been incorporated insofar as the apatite formation in SBF is maintained [34,54]. Based on these reports, the bone-bonding capacity and cytotoxicity of the treated Ti and its alloy samples in the present study were assessed in terms of apatite formation in SBF and in cell culture, respectively.

To incorporate iodine into the calcium titanate, the Ti and alloys with calcium titanate were soaked in four types of iodine solutions that comprised ICl_3_, ICl, PVP-I, and NaI. The former two produced positively charged iodine ions in an aqueous solution, whereas the latter two produced isolated and/or negatively charged iodine ions. As shown in Table 2, iodine was incorporated into the metal surfaces only when the metals were soaked in ICl_3_ and ICl. This indicates that iodine ions such as I^+^ and I^3+^ were incorporated by ion exchange with positively charged calcium ions in the calcium titanate. This is consistent with the results in Table 2, where the Ti with the lower calcium content has a higher iodine content. The narrow XPS spectra in Figure 1a show that positively charged iodine was detected along with isolated or negatively charged iodine. Iodine is easily oxidizable and reduces halogen, and is stable when isolated or in single negative ion form in aqueous solution at neutral pH.

In this study, positively charged ions were incorporated into the calcium titanate that formed on Ti and its alloys by ion exchange, as described above. A portion transformed into stable isolated ions or single negative ion as the result of absorbing electrons during the washing process after the solution treatment and/or the XPS measurement (Appendix A). The formed negatively charged iodine ions might be present on the grain boundaries and surfaces of the calcium titanate and combine with calcium ions in the calcium titanate. According to TF-XRD, a slight shift in the calcium titanate peak at 25.4° to 24.8° in 2θ was detected, indicating that the d space of the calcium titanate expanded from 3.204 to 3.587 due to the exchange of calcium ions by iodine ions. Although information is scarce on the ionic radius of I^+^ and I^3+^, they should be between 1.40 Å atomic radius and 1.09 Å ionic radius of I^7+^, which are larger than Ca^2+^ (0.99 Å) and can cause the above increase in d space. Detailed analysis with Rietveld refinement was, however, difficult because of the broad peak. Further studies such as TEM equipped with ED are needed to determine the precise crystal structure as well as the interplanar distance of the calcium titanate. Notably, a large amount or absence of iodine incorporation was observed again when the Ti with sodium hydrogen titanate produced by the NaOH treatment was soaked in ICl_3_ or PVP-I solution, respectively (Appendix A), because of its layered structure of sodium hydrogen titanate, which is similar to calcium titanate [52]. However, the critical scratch load of the treated Ti was as low as 5.5 mN. Subsequent heat treatment increased the value to 34.7 mN, but the iodine vanished from the metal surface due to the low boiling point of iodine (~184 °C). These results suggest that an iodine-loaded titanate surface layer with a high scratch resistance can only be formed when the iodine solution treatment is performed after the heat treatment.

Figure 2 shows that the surface layer tended to be damaged by soaking in the ICl_3_ solution at the high concentration of 100 mM, especially at an elevated temperature such as 80 °C. As a result, the moderate concentration of 10 mM ICl_3_ incorporated the maximum amount of iodine into the surfaces of the Ti and alloys, with the amounts being 8.6%, 10.5%, and 7.3% for Ti, Ti-6Al-4V, and Ti-15Zr-4Nb-4Ta, respectively. The damage on the surface layer might also have affected the scratch resistance of the samples. Although statistically significant differences were not detected, all the average values of the scratch resistance on the Ti samples treated with 100 mM ICl_3_ at 80, 60, and 40 °C were slightly lower than that on Ti treated with 10 mM ICl_3_ (Table 3). Table 3 also shows that Ti-15Zr-4Nb-4Ta treated with 10 mM ICl_3_ with the detectable alloying elements of Zr, Nb, and Ta incorporated in its surface exhibited significantly greater scratch resistance compared with the Ti subjected to the same treatment, whereas the Ti-6Al-4V without any detectable alloying elements showed a comparable value. This suggests that residual alloying elements in the surface layer may affect the scratch resistance of the metal by forming some types of metal oxides with superior mechanical hardness such as ZrO_2_ and Nb_2_O_5_.

Zeta potential titration curve measurement revealed that the treated metal exhibited an IEP shift from 4.1 for the untreated Ti to 5.3 and both an acid and a basic plateau, attributable to the presence of both basic and acidic Ti-OH groups (Figure 4). The presence of abundant Ti-OH (both basic and acidic) groups was confirmed in the O1s fitting profile obtained by XPS analysis (Figure 1c). This is in contrast to the calcium titanate treated with hot water, which exhibited no IEP shift and no evident plateau [55]. This indicates that ICl_3_ more effectively attacks the calcium titanate than hot water to form surface functional groups that act as a strong base (i.e., able to be protonated at a relatively high pH). When the zeta potential of the treated Ti was measured in a diluted SBF electrolyte, it initially showed a negative zeta potential of approximately −38 mV (i.e., close to the one measured in KCl), which gradually increased to −33 mV (Figure 7a). The increase in the zeta potential is probably due to the adsorption of calcium ions from the SBF due to the surface bioactivity [55]. A similar increase was observed on the highly reactive bioactive glass, but was not evident on the calcium titanate subjected to hot water treatment [55]. Conversely, the pH of the SBF gradually decreased with time. This can be explained by considering the release of positively charged iodine ions from the treated surface that is unstable at physiological pH, which may form HIO by consuming the OH^−^ ions in the surrounding solution [56]. As a result, the treated Ti and its alloys formed apatite in SBF within 3 days (Figure 5 and Figure 6). This apatite formation can probably be attributed to the abundant Ti-OH groups as well as the release of Ca ions from the treated metals. The treated Ti and its alloys are thus expected to bond to living bone because of their apatite formation capability [15,16,57]. The bone-bonding capability of the treated metals is under study using animal model and will be reported in the future.

It was reported that both isolated and negatively charged iodine can be incorporated into the surface of aluminum Ti and Ti-6Al-4V, after the metals were anodized to form micrometer-/submicrometer-scale surface pores, and were subsequently subjected to electrodeposition treatment in PVP-I solution [38,39,40]. The treated metals not only exhibited in vitro antibacterial activity against *S. aureus* and *E. coli*, but also clinically prevented infection for 158 patients including three tumor cases that developed acute infection, but recovered without removal of the implants. Additionally, 64 patients with infection were cured using the iodine-loaded implants. However, the amount of iodine was limited to a range of 0.1% to 1%, since iodine can be adsorbed only on the inner walls of the pores formed by anodic oxidation. In this study, up to 10.5% iodine was incorporated into Ti and its alloys using the layered crystalline structure of calcium titanate. It is expected that the iodine-loaded Ti developed in this study will be able to prevent the occurrence of postoperative infection to the same degree or higher because of its higher amount of iodine. The cell culture shown in Figure 9 demonstrates that the metal with high iodine content displayed no cytotoxicity in MC3T3-E1 and L929 cells. At day 3, a statistically significant increase was detected for MC3T3-E1. This might be attributed to the nanometer-scale surface roughness of the treated metal with the iodine-containing calcium-deficient calcium titanate. The increased proliferation of primary mouse calvarial osteoblasts on a similar nanostructured surface of Ti with sodium titanate was reported [58]. By contrast, a statistically significant decrease was detected for L929 at day 3. This might be attributable to iodine ions released from the treated metal. Although there are few studies on the cytotoxicity of iodine, James et al. reported that the cell viability of human fibroblasts, osteoblasts, and myoblasts was unaffected by exposure to PVP-I at up to 100 ppm but decreased to less than 10% by exposure to over 1000 ppm of PVP-I [59]. In this study, the released concentration of iodine reached up to 5.6 ppm within 90 days, a level lower than the concentration suggested to cause cytotoxicity. Despite the relatively lower concentration of iodine ions, they tend to accumulate near the surface and disturb the initial cell proliferation of L929. This local accumulation is, however, reduced in vivo because of the surrounding dynamic blood flow. Brown et al. showed that borate bioactive glasses with higher B_2_O_3_ content resulted in greater inhibition of MC3T3-E1 cell proliferation under a static cell culture condition, whereas no cytotoxicity was observed under dynamic culture conditions mimicking the body environment [60]. Rahaman et al. reported that these borate bioactive glasses exhibit no observable toxic effects in vivo [61]. At day 7, no statistically significant differences were detected for either MC3TC-E1 or L929, even in the static culture condition, in this study. When the Ti with the 8.6% and 2.3% iodine was subjected to the antibacterial test, both exhibited potent activity against MRSA, as shown in Figure 10. Although the former antibacterial activity was almost maintained at the same level for 1 week, even after the humidity and water resistance tests, the latter decreased after the same tests, indicating the lower stability of the antibacterial activity for the Ti with the lower iodine content. The water resistance test for up to 6 months indicated that sustainable antibacterial activity can be conferred on the Ti with 8.6% iodine. The treated Ti also exhibited strong antibacterial activity (reduction rate > 99%), even for *S. aureus*, *E. coli*, and *S. epidermidis*, suggesting a broad antibacterial spectrum. It was reported that surgical site infections in cases of artificial joint replacement are mainly caused by MRSA (42%), followed by *S. aureus* (17%), *S. epidermidis* (11%), *E. coli* (4%), and others [62].

Based on these results, we expect that Ti and its alloys treated with 10 mM ICl_3_ at 80 °C following NaOH-CaCl_2_ heat treatment prevent infection in most cases while also bonding to living bone due to their apatite-formation capability.

The limitation in this study is that the capacities of bone formation and antibacterial activity were evaluated in vitro; therefore, in vivo studies are needed to prove the benefit of the iodine-loaded Ti and its alloys. A more detailed cell study is also required to reveal cell–material interactions and the effects of the treated metals on gene expression.

## 5. Conclusions

A novel solution and heat treatment that introduces a large amount of positively-charged iodine ions into Ti and its alloy surfaces were developed through an ion-exchange reaction using the layered structure of calcium titanate. The iodine-containing calcium-deficient calcium titanate that was thus formed slowly released 5.6 ppm of iodine ions over 90 days. The treatment confers the following characteristics on Ti and its alloys, such as Ti-6Al-4V and Ti-15Zr-4Nb-4Ta:1.High-capacity apatite formation: The treated metals have abundant acidic and basic Ti-OH groups and form apatite in SBF within 3 days;2.Cytocompatibility: The Ti loaded with 8.6% iodine did not exhibit cytotoxicity to MC3T3-E1 or L929 cells;3.Broad antibacterial spectrum: The Ti loaded with 8.6% iodine exhibited potent antibacterial activity (reduction rate > 99%) against not only MRSA but also *S. aureus*, *E. coli*, and *S. epidermidis*;4.Sustainable antibacterial activity: The Ti loaded with 8.6% iodine exhibited a 97.3% reduction in MRSA even after being soaked in PBS under physiological conditions for 6 months.

All these functions are critical for patients with infected implants. Ti and its alloys with multifunctional surface layers will be particularly useful for orthopedic and dental implants since they reliably bond to bone and are able to help prevent infection owing to their apatite formation, cytocompatibility, and sustainable antibacterial activity.

## Figures and Tables

**Figure 1 nanomaterials-11-02199-f001:**
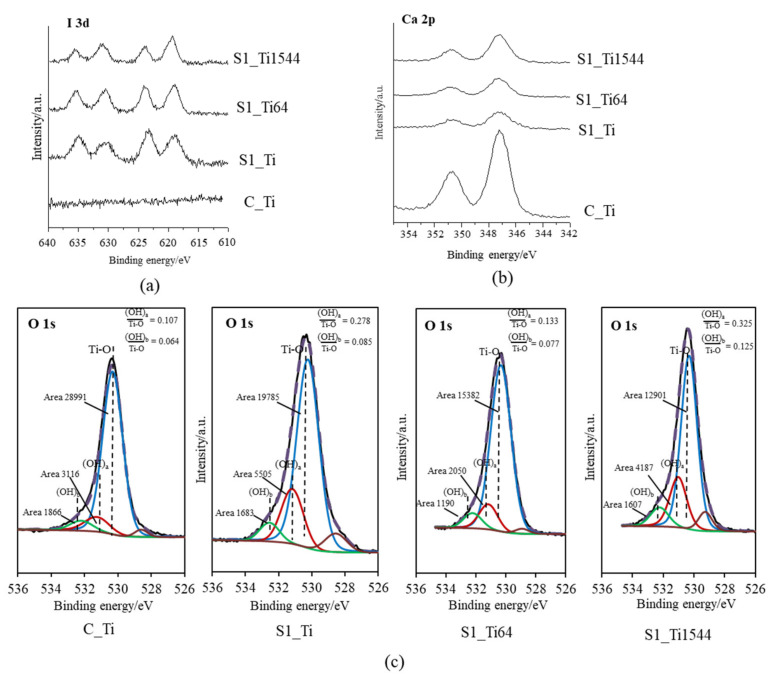
XPS profiles of (**a**) I 3d, (**b**) Ca 2p on Ti and its alloys soaked in 10 mM ICl_3_ at 80 °C following NaOH-CaCl_2_ heat treatment, and (**c**) their O 1s fitting profiles.

**Figure 2 nanomaterials-11-02199-f002:**
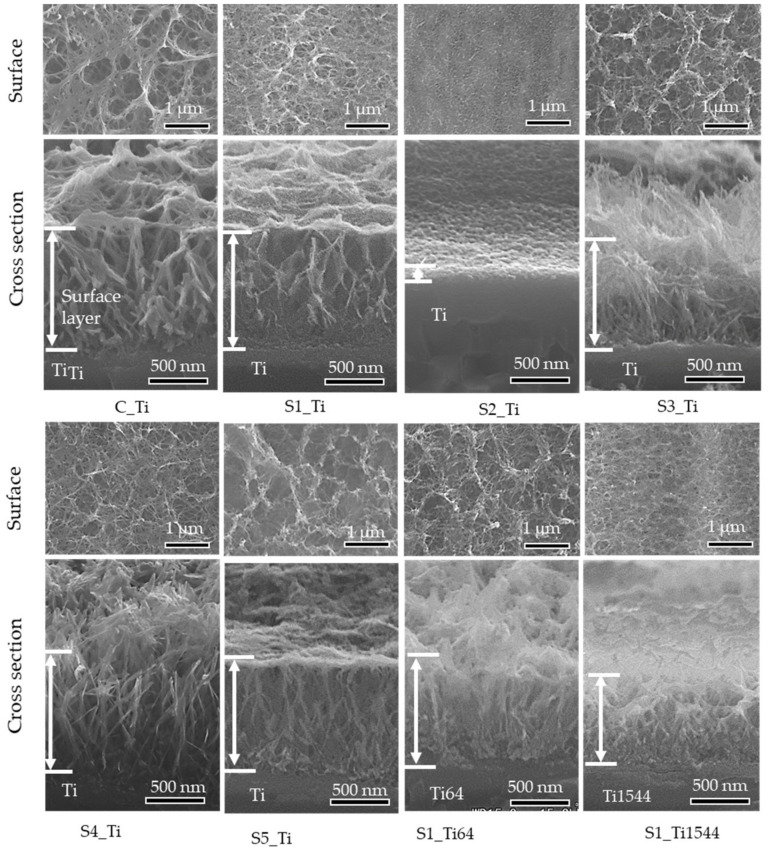
Surface and cross-sectional FE-SEM images of Ti and its alloys soaked in ICl_3_ or ICl solution.

**Figure 3 nanomaterials-11-02199-f003:**
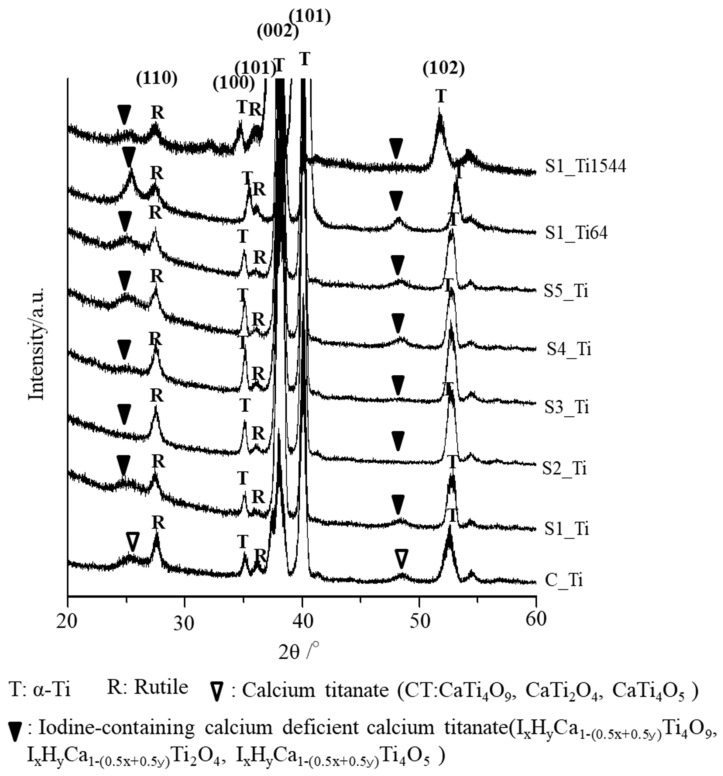
TF-XRD profiles of Ti and its alloys soaked in ICl_3_ or ICl solution following NaOH-CaCl_2_ heat treatment.

**Figure 4 nanomaterials-11-02199-f004:**
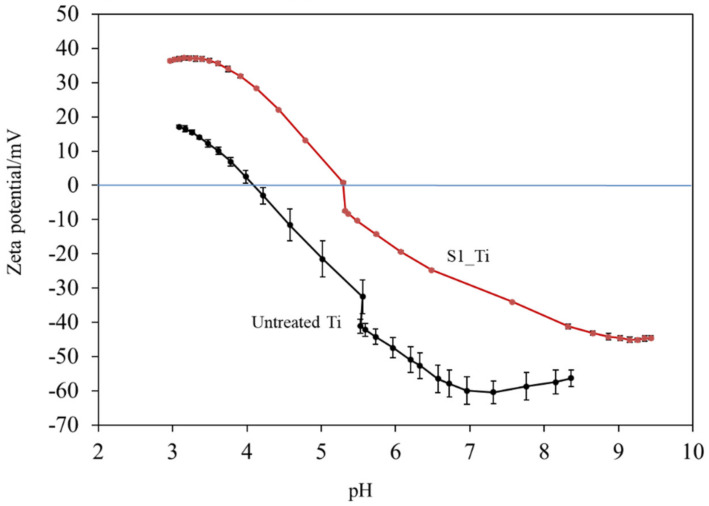
Zeta potential titration curve for Ti untreated and treated with 10 mM ICl_3_ at 80 °C following NaOH-CaCl_2_ heat treatment.

**Figure 5 nanomaterials-11-02199-f005:**
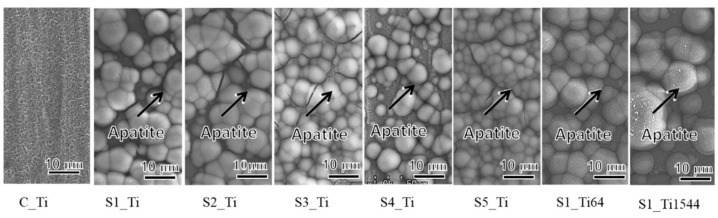
FE-SEM images of the surfaces of Ti and its alloy soaked in SBF for 3 days after ICl_3_ or ICl solution treatment following NaOH-CaCl_2_ heat treatment.

**Figure 6 nanomaterials-11-02199-f006:**
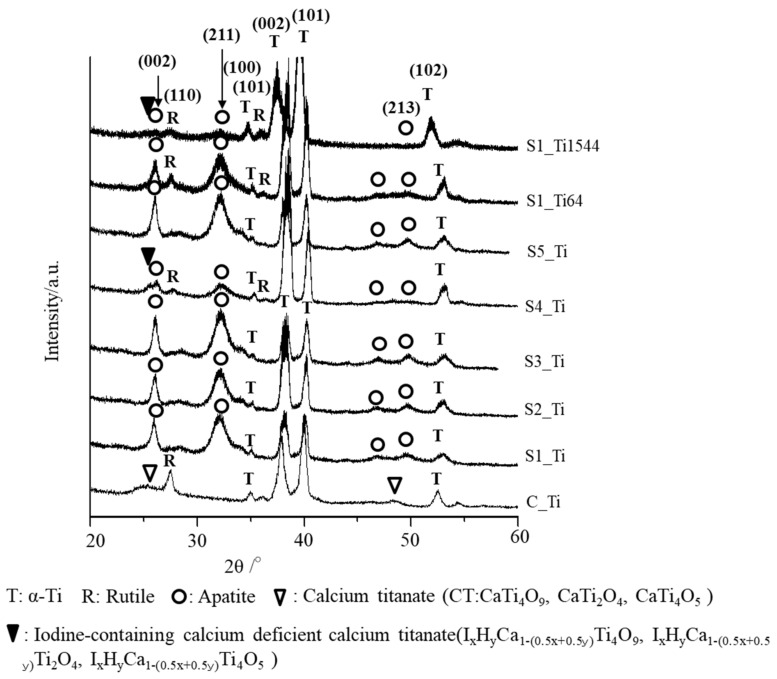
TF-XRD profiles of the surfaces of Ti and its alloy soaked in SBF for 3 days after ICl_3_ or ICl solution treatment following NaOH-CaCl_2_ heat treatment.

**Figure 7 nanomaterials-11-02199-f007:**
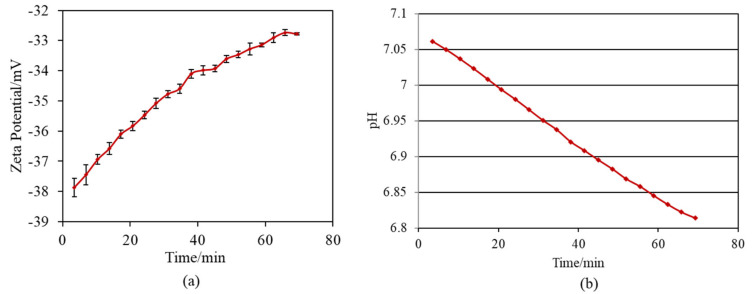
Variation per minute of (**a**) zeta potential of sample and (**b**) pH of SBF during soaking S1_Ti in diluted SBF.

**Figure 8 nanomaterials-11-02199-f008:**
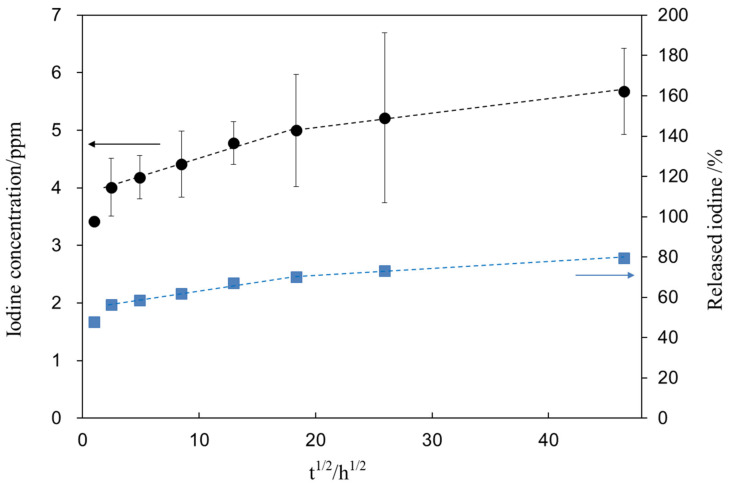
Iodine concentration released from S1_Ti as a function of the square root of soaking time in PBS. The percentage of released iodine from S1_Ti was calculated from the iodine concentration in PBS and iodine amount remaining on the metal after the ion release test.

**Figure 9 nanomaterials-11-02199-f009:**
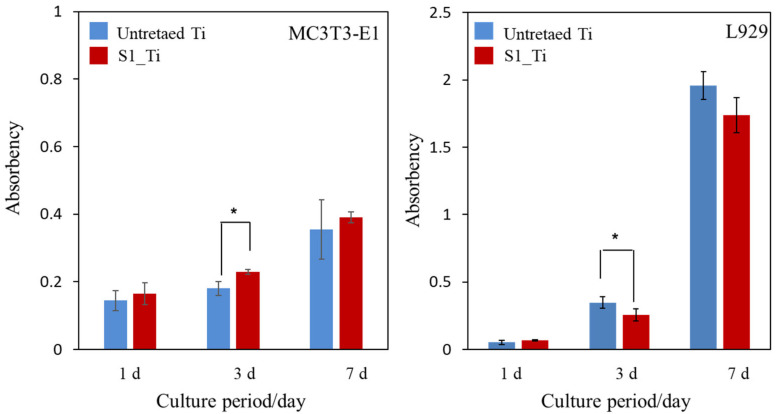
Viability of cells cultured on untreated Ti and S1_Ti; * *p* < 0.05.

**Figure 10 nanomaterials-11-02199-f010:**
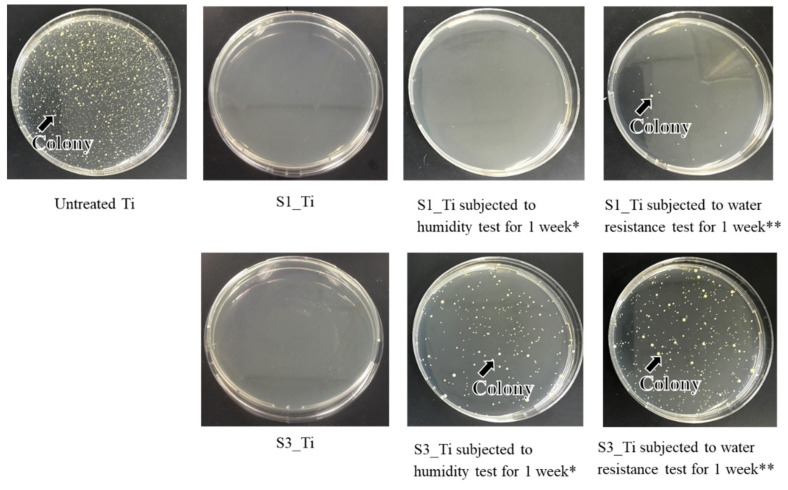
Optical images of colony formation on untreated Ti, S1_Ti, and S3_Ti with or without short-term stability test; * Samples were retained under a temperature of 80 °C and relative humidity of 95% for 1 week; ** Samples were retained in phosphate-buffered saline at 36.5 °C for 1 week while shaking at a rate of 50 strokes/min.

**Table 1 nanomaterials-11-02199-t001:** Notations of Ti and its alloy samples used in this study.

Sample	Substrate	NaOH-CaCl_2_-Heat Treatment	Iodine Treatment
Untreated Ti	Ti	Not treated	Not treated
C_Ti	Ti	Treated	Not treated
S1_Ti	Ti	Treated	10 mM ICl_3_, 80 °C
S2_Ti	Ti	Treated	100 mM ICl_3_, 80 °C
S3_Ti	Ti	Treated	100 mM ICl_3_, 60 °C
S4_Ti	Ti	Treated	100 mM ICl_3_, 40 °C
S5_Ti	Ti	Treated	10% ICl, 60 °C
S1_Ti64	Ti-6Al-4V	Treated	100 mM ICl_3_, 80 °C
S1_Ti1544	Ti-15Zr-4Nb-4Ta	Treated	100 mM ICl_3_, 80 °C
S6_Ti	Ti	Treated	1000 ppm Polyvinylpyrrolidone, 60 °C
S7_Ti	Ti	Treated	100 mM NaI, 60 °C

**Table 2 nanomaterials-11-02199-t002:** The results of quantitative analysis of XPS on Ti and its alloys soaked in ICl_3_, ICl, or PVP-I solution following NaOH-CaCl_2_ heat treatment.

	Element/Mass%								
Sample	O	Ti	Al	V	Zr	Nb	Ta	Ca	I
Untreated Ti	48.6	51.4	-	-	-	-	-	0.0	0.0
C_Ti	43.2	48.5	-	-	-	-	-	8.3	0.0
S1_Ti	42.7	46.1	-	-	-	-	-	2.6	8.6
S2_Ti	41.2	55.4	-	-	-	-	-	1.4	1.9
S3_Ti	39.0	52.4	-	-	-	-	-	6.2	2.3
S4_Ti	39.7	52.1	-	-	-	-	-	7.5	0.7
S5_Ti	44.2	50.5	-	-	-	-	-	2.3	3.1
S1_Ti64	40.0	46.3	0	0	-	-	-	3.2	10.5
S1_Ti1544	37.1	39.3	-	-	3.0	1.9	7.3	4.0	7.3
S6_Ti	48.0	45.4	-	-	-	-	-	6.6	0.0
S7_Ti	43.1	50.2	-	-	-	-	-	6.8	0.0

**Table 3 nanomaterials-11-02199-t003:** Scratch resistance of Ti and its alloys soaked in ICl_3_ or ICl solution following; NaOH-CaCl_2_ heat treatment.

Sample	Critical Scratch Load (mN)	Statistically Significant Difference
C_Ti	31.8 ± 2.0	†
S1_Ti	33.3 ± 8.8	†
S2_Ti	27.6 ± 2.9	†,*
S3_Ti	25.3 ± 5.0	†,*
S4_Ti	26.5 ± 2.5	†,*
S5_Ti	26.4 ± 2.8	†,*
S1_Ti64	41.5 ± 4.2	†
S1_Ti1544	119.0 ± 9.9	

†: Statistical significant difference compared to S1_Ti1544 (*p* < 0.05); *: Statistical significant difference toward S1_Ti64 (*p* < 0.05).

**Table 4 nanomaterials-11-02199-t004:** Characteristics of Ti and its alloys subjected to NaOH-CaCl_2_ heat treatment and subsequent iodine treatment.

Sample	Ca (Mass%)	I (Mass%)	Crystallin Phase *	Critical Scratch Load (mN)	Apatite Formation (Soaked in SBF for 3 d)
C_Ti	8.3	0.0	CT, R	33.3 ± 2.0	×
S1_Ti	2.6	8.6	ICT, R	33.3 ± 8.8	○
S2_Ti	1.4	1.9	ICT, R	27.6 ± 2.9	○
S3_Ti	6.2	2.3	ICT, R	25.3 ± 5.0	○
S4_Ti	7.5	0.7	ICT, R	26.5 ± 2.5	○
S5_Ti	2.3	3.1	ICT, R	26.4 ± 2.8	○
S1_Ti64	3.2	10.5	ICT, R	41.5 ± 4.2	○
S1_Ti1544	4.0	7.3	ICT, R	119.0 ± 9.9	○
S6_Ti	6.6	0.0	**	-	-
S7_Ti	6.8	0.0	-	-	-

* R: Rutile; **: not investigated; CT: Calcium titanate; ICT: Iodine-containing calcium deficient calcium titanate. ×: no apatite formation, ○: apatite formed

**Table 5 nanomaterials-11-02199-t005:** Antibacterial activity value of S1_Ti against *MRSA* and *S. aureus* and its short-term stability.

	Average of *MRSA* Count/CFU			Average of *S. aureus* Count/CFU		
Sample	After Inoculation	After Incubation	A Value *	After Inoculation	After Incubation	A Value
Untreated Ti	2.3 × 10^6^	4.4 × 10^5^	-	1.2 × 10^7^	6.0 × 10^6^	-
S1_Ti	1.3 × 10^6^	<20	4.3	1.1 × 10^7^	<20	5.5
S1_Ti + MR ** (1 week)	8.0 × 10^5^	<20	4.3	1.0 × 10^7^	1.2 × 10^3^	3.7
S1_Ti + WR *** (1 week)	1.9 × 10^6^	2.2 × 10^3^	2.3	1.2 × 10^7^	1.3 × 10^5^	1.7

** MR: humidity test; *** WR: water resistance test; * A value: antibacterial activity value.

**Table 6 nanomaterials-11-02199-t006:** Antibacterial activity value of S1_Ti against *S.*
*e**pidermidis* and *E. c**oli* and its shor- term stability.

	Average of *E. coli* Count/CFU			Average of *S. epidermidis* Count/CFU		
Sample	After Inoculation	After Incubation	A Value *	After Inoculation	After Incubation	A Value
Untreated	1.6 × 10^5^	1.7 × 10^6^	-	4.0 × 10⁴	8.5 × 10⁴	-
S1_Ti	1.2 × 10^5^	<20	4.9	4.1 × 10⁴	<20	3.6
S1_Ti + MR ** (1 week)	1.2 × 10^5^	<20	4.9			
S1_Ti + WR *** (1 week)	8.8 × 10^4^	9.2 × 10^3^	2.3			

** MR: humidity test, *** WR: water resistance test, * A value: antibacterial activity value.

## Data Availability

The data presented in this study are available upon request from the corresponding author.

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
