# Peer review of "Iodine-Loaded Calcium Titanate for Bone Repair with Sustainable Antibacterial Activity Prepared by Solution and Heat Treatment"

_nanomaterials, 2021, doi:10.3390/nano11092199_

Round 1

Reviewer 1 Report

Comments to Authors on the Manuscript Number: 1325856

The paper “Iodine-Carrying Titanium for Bone Repair with Sustainable Antibacterial Activity Prepared by Solution and Heat Treatment”, by S. Yamaguchi, P.T.M. Le, S.A. Shintani, H. Takadama, M. Ito, S. Ferraris and S. Spriano, approaches the development of a series of layered structures based on titanium and its alloys, which were first coated with calcium titanate and then modified by immersion in iodine-containing solutions and heat treatment. The obtained samples were characterized from multiple points of view in order to demonstrate their suitability for bone substitutes having bioactive, biocompatible and antibacterial properties.

The article presents several weaknesses, as follows:

  1. Title: - I suggest changing the phrase “iodine-carrying titanium” with a more appropriate one since the bonds are of chemical nature and iodine is not incorporated in the metal.
  2. Abstract: - the first part needs to be slightly revised because the meaning is affected by the phrases topic.
  3. Keywords: - I suggest replacing the phrase “apatite formation” with a more eloquent word or phrase.
  4. Materials and methods: - I suggest including a table with the names and characteristics of the produced samples.
  5. Results: - the samples names are too long (suggestive short codes should be employed);

- how were the samples prepared for the cross-section evaluation through SEM?

- some references are necessary when proposing the formation of iodine-containing calcium titanates;

- the shifts occurred in the case of the XRD patterns must be better highlighted by plotting in a different way;

- I suggest including the codes of the employed database sheets of different crystalline compounds, as well as the Miller indices in the figures presenting the XRD patterns;

- Figure 4 does not have a legend;

- the Ox scale of Figure 6 should be established from 20 to 60 °; enlarged areas from the XRD patterns must be presented separately in order to sustain the comments.

  1. Discussion: - how was the inter-planar distance in calcium titanate calculated? a Rietveld refinement analysis would be more reliable;

- what solution do the authors propose for improving the biological response in the presence of different types of cells?

  1. Conclusions: - I suggest reconsidering the first part since it is confusing;

- which are the perspectives and limitations?

  1. References: - more recent studies are required.
  2. The whole manuscript must be revised by a native speaker since there are many problems related to the language.

In conclusion, the paper “Iodine-Carrying Titanium for Bone Repair with Sustainable Antibacterial Activity Prepared by Solution and Heat Treatment”, by S. Yamaguchi, P.T.M. Le, S.A. Shintani, H. Takadama, M. Ito, S. Ferraris and S. Spriano, can be published in Nanomaterials (MDPI) after a major revision.

Author Response

Please find the author's reply in the attachment.

Reviewer 2 Report

Authors presents a novel antibacterial methods for titanium implants based on ionided. The amount of experiments is good and results are promising.

As suggestions, this reviewer recommends:

  • Figure 2: increase the size of each micrograph and add the scalebar to the one on the right.
  • Add the supplementary info to the manuscript.

Author Response

(The authors gave the same response as above.)

Reviewer 3 Report

This paper presents iodine-carrying titanium for bone repair with sustainable anti-bacterial activity. To make advanced Ti implants  with  simultaneously antibacterial activity and bone-bonding capacity is still an outstanding challenge in the orthopedic and dental fields.  This paper presents interesting approach by coating Ti surface with melted iodine which is well known to have strong antibacterial properties. This is new approach with valuable contribution to this field and solving significant problem. The paper is well presented and it is recommended to be published in this journal. There are few comments for authors to address to further improve the quality of their paper.

1.In experimental sections the section of used chemicals and their sources is missing

2.In table 1 there is some inconsistency that lower conc. 10mM ICI that make less amount of I on surface compared when higher conc 100mMICI was used with the same conditions (80C)

3.Figure 2 are not clearly presenting structural features and should be improved, making images wider, removing arrows and making smaller fonts.   

4.Fig.4 it is not clear which curve is treated or not, label on graph or explain in captions.

5.Results with iodine loading (mass) on Ti substrates should be provided

6.Figure 10. The release kinetic of Iodine should include other graph conc vs the time and % of released vs time which are more practically relevant. To have kinetic rate of I will be very valuable.  This section is expected to be presented before cell/bacteria study

Author Response

(The authors gave the same response as above.)

Round 2

Reviewer 1 Report

The authors have invested efforts to respond to the reviewer's comments and improve the manuscript, which leads to the conclusion that the paper can be publushed in Nanomaterials in its current form.

Reviewer 3 Report

Authors satisfactory answered  all questions and improved their MS